# Dietary and Nutrition Interventions for Breast Cancer Survivors: An Umbrella Review

**DOI:** 10.3390/nu18010030

**Published:** 2025-12-21

**Authors:** Joan Ern Xin Tan, Mattias Wei Ren Kon, Charmaine Su Min Tan, Kevin Xiang Zhou, Kewin Tien Ho Siah, Serene Si Ning Goh, Qin Xiang Ng

**Affiliations:** 1NUS Yong Loo Lin School of Medicine, National University of Singapore, 10 Medical Dr, Singapore 117597, Singaporemattias.kon@u.nus.edu (M.W.R.K.);; 2Faculty of Medicine & Health Sciences, McGill University, Montreal, QC H3G 2M1, Canada; 3Division of Gastroenterology & Hepatology, University Medicine Cluster, National University Hospital, Kent Ridge, Singapore 119077, Singapore; 4Saw Swee Hock School of Public Health, National University of Singapore, 12 Science Drive 2, #10-01, Singapore 117549, Singapore; 5Department of General Surgery, National University Hospital Singapore, Kent Ridge, Singapore 119077, Singapore

**Keywords:** breast cancer, survivorship, dietary intervention, nutrition, Mediterranean diet, quality of life, umbrella review

## Abstract

Background/Objectives: Breast cancer is the most common malignancy among women globally, with survival rates improving due to earlier detection and better treatment. As a result, cancer survivors now constitute a growing segment of the population, and addressing their long-term health and well-being is a public health priority. Diet and nutrition represent modifiable factors that may influence recurrence, comorbidities, and quality of life (QoL), yet clear evidence-based guidance remains limited. This umbrella review thus synthesized evidence from published reviews on the effects of dietary and nutrition interventions among breast cancer survivors. Methods: Following a prospectively registered protocol in PROSPERO (CRD420251185022), six databases (PubMed, EMBASE, Scopus, Cochrane Library, PsycINFO and CINAHL) were systematically searched for systematic reviews/meta-analyses evaluating dietary or nutrition interventions in adult breast cancer survivors. Eligible reviews reported anthropometric, metabolic, psychosocial, or survival outcomes. Methodological quality was appraised using the AMSTAR-2 tool, and findings were narratively synthesized. Results: Nine systematic reviews encompassing more than 10,000 breast cancer survivors were included. Interventions ranged from general dietary counselling and structured weight-management programmes to Mediterranean-style dietary patterns, dietitian-led primary care, multiple health behaviour change interventions, mobile nutrition apps, and broader lifestyle programmes incorporating diet. Across reviews, interventions consistently improved diet quality and fruit–vegetable intake, produced modest but meaningful reductions in weight, body mass index, and body fat, and enhanced several QoL domains (e.g., fatigue, physical functioning, body image). Higher adherence to Mediterranean-style diets was associated with lower all-cause and non–breast cancer mortality, though certainty was limited by observational designs. However, evidence for long-term maintenance, survival endpoints, and ethnically diverse or low- and middle-income populations remains sparse. Conclusions: Dietary and nutrition interventions, particularly structured, dietitian-supported, and Mediterranean-style approaches, contribute to improved diet quality, sustainable weight control, and enhanced QoL among breast cancer survivors. Integrating nutrition care into survivorship pathways should be the focus of future research.

## 1. Introduction

Breast cancer remains the most frequently diagnosed malignancy among women worldwide and a leading cause of cancer-related mortality. In 2022 alone, it was estimated that 2.31 million new cases of breast cancer were diagnosed worldwide, comprising 11.6% of all cancers [1].

Advances in screening, early detection, and multimodal therapy have contributed to a 43% reduction in mortality and a growing population of long-term survivors [2]. As of 2023, more than 4 million breast cancer survivors live in the United States [3]. This demographic shift compels a paradigm change in oncology, from focusing solely on treatment to optimizing survivorship, recovery, and quality of life [4].

Cancer survivors form a substantial part of our society, and focusing on their needs and expectations has become increasingly important. Diet and nutrition are two domains in which survivors often seek guidance, yet evidence-based recommendations remain underdeveloped. Although the role of diet and nutrition in the primary prevention of cancer has been well established and formal dietary guidelines exist for the general population, such targeted, evidence-based recommendations for cancer survivors are lacking [5]. More methodologically rigorous epidemiological studies are therefore required to clarify the influence of diet, ranging from nutrient-level intake to whole-dietary patterns, on survival and broader health outcomes among survivors.

The transition from active treatment to survivorship is often accompanied by persistent physical and psychosocial challenges, including fatigue, weight changes, menopausal symptoms, and increased cardiometabolic risk [6,7]. In discrete choice experiments, it has been found that patients themselves, particularly younger women and those with early-stage disease, prioritized quality of life (QoL), reduced toxicity and treatment burden over modest survival gains value [8]. Addressing these sequelae requires long-term management strategies aimed at mitigating treatment-related toxicity, preventing recurrence, and promoting overall well-being. Among modifiable health behaviours, diet and nutrition have emerged as key determinants that can influence survival, recurrence, and quality of life [9,10,11].

Discrete choice experiments in breast cancer have consistently shown that many survivors, particularly younger women and those with early-stage disease, prioritise quality of life, reduced treatment toxicity, and lower treatment burden over small absolute gains in survival, highlighting the importance of supportive interventions such as nutrition care in survivorship planning [12,13,14]. Yet the literature spans multiple intervention modalities—from general dietary counselling and calorie restriction to Mediterranean-style eating and structured dietetic follow-up—each with varying levels of methodological quality and clinical relevance. Given this complexity, an umbrella review, which consolidates and critically appraises existing systematic reviews and meta-analyses [15], provides a rigorous means of synthesizing evidence and identifying consistent effects and research gaps.

Accordingly, this umbrella review aims to (1) synthesize evidence from systematic reviews and meta-analyses evaluating dietary and nutrition interventions in breast cancer survivors, (2) examine their effects on anthropometric, metabolic, psychosocial, and survival outcomes, and (3) identify implications for practice, policy, and future research in breast survivorship care.

## 2. Methods

### 2.1. Review Protocol

This umbrella review (or overview of systematic reviews and meta-analyses) aimed to synthesize and critically appraise evidence on dietary and nutrition interventions in breast cancer survivorship. The review methodology was informed by the Cochrane Handbook for Umbrella Reviews [16] and followed the reporting guidelines by Preferred Reporting Items for Systematic Reviews and Meta-Analyses (PRISMA) 2020 [17]. The protocol was also prospectively registered with the International Prospective Register of Systematic Reviews (PROSPERO) (registration number: CRD420251185022).

### 2.2. Search Strategy

To identify relevant systematic reviews and meta-analyses, comprehensive searches were performed across PubMed, Embase, CINAHL, PsycINFO, Scopus, and the Cochrane Library. Search terms combined controlled vocabulary (e.g., “breast neoplasms,” “diet therapy,” “nutrition counseling,” “dietitian”) with free-text keywords (“survivor*,” “weight management,” “Mediterranean diet,” “quality of life”). The detailed search strategy for each individual database can be found in the Appendix A. Searches were restricted to English-language publications from database inception up to end July 2025. Manual reference screening of included reviews and citation chaining in Google Scholar were performed to ensure completeness. Grey literature and non-peer-reviewed sources were excluded.

### 2.3. Eligibility Criteria

Systematic reviews and meta-analyses were included if they: (1) Explicitly described systematic methods for literature identification, selection, and synthesis; (2) Focused on adult (≥18 years) breast cancer survivors who had completed primary treatment (surgery, radiotherapy, and/or chemotherapy); (3) Evaluated dietary or nutrition interventions, including dietary counseling, weight-management programs, Mediterranean or plant-based dietary patterns, or structured dietetic care; (4) Reported health-related outcomes such as anthropometry (weight, body mass index (BMI), waist circumference), metabolic biomarkers, recurrence, mortality, or quality of life; and (5) Included randomized controlled trials (RCTs), prospective cohorts, or other controlled comparative designs.

Reviews that exclusively examined non-breast-cancer populations, non-interventional designs, or non-dietary supportive care (e.g., physical activity only) were excluded unless breast-cancer-specific subanalyses were reported.

Primary outcomes of interest include: anthropometric measures (e.g., body weight, BMI, waist circumference, body-fat percentage), metabolic indicators (e.g., lipid profiles, glucose, insulin resistance, inflammatory biomarkers), psychosocial and QoL outcomes: global and domain-specific QoL (e.g., EORTC-QLQ-C30, FACT-B), and survival outcomes (recurrence, cancer-specific and all-cause mortality).

### 2.4. Screening and Selection Process

All records retrieved were imported into Covidence (Veritas Health Innovation, Melbourne, Australia) for screening and deduplication. Three independent investigators (J.E.X.T., M.W.R.K. and C.S.M.T.) screened titles, abstracts, and full texts in duplicate using the inclusion criteria. Discrepancies were resolved through discussion and adjudication by a senior investigator (Q.X.N. or S.S.N.G.) to ensure consistency.

### 2.5. Data Extraction

A standardized extraction template was used to capture review characteristics (author, year, databases searched, number and type of primary studies, intervention type, comparators, outcomes, key findings, and quality-appraisal methods). Each included review was independently extracted by two reviewers (J.E.X.T. and M.W.R.K.) and cross-checked by a third (C.S.M.T.) to ensure accuracy.

### 2.6. Quality Appraisal and Certainty of Evidence

Methodological quality of the included systematic reviews was evaluated using AMSTAR-2 (A Measurement Tool to Assess Systematic Reviews, version 2) [18], focusing on key domains such as protocol registration, search comprehensiveness, risk-of-bias assessment, and appropriateness of synthesis. Each review was independently rated by three reviewers, with disagreements resolved by consensus with the senior author. Certainty of evidence for key outcomes was summarized according to the Grading of Recommendations, Assessment, Development and Evaluation (GRADE) approach [19], as reported by the original reviews.

### 2.7. Data Synthesis

Given the heterogeneity in intervention types and outcome measures, a narrative synthesis approach was used following the framework of Popay et al. [20]. Data were organized by thematic domains, namely, general dietary interventions, weight-management programs, Mediterranean diet adherence, diet quality and QoL, and dietetic care. Quantitative results (e.g., pooled mean differences, hazard ratios) from included meta-analyses were extracted verbatim to illustrate magnitude and direction of effects. Findings were summarized descriptively and supported by GRADE evidence tables.

To determine the degree of overlap between the included papers, one study has suggested the use of corrected covered area (CCA) index [21]. The overlapping primary studies were analysed and have been depicted in Appendix A.

### 2.8. Ethical Approval

No ethics approval is necessary as this is a review of published studies and no human participants were directly involved.

## 3. Results

### 3.1. Literature Retrieval

As illustrated in Figure 1, from 902 records, five systematic reviews and meta-analyses met the inclusion criteria, collectively encompassing more than 10,000 women with breast cancer who had completed active treatment. The reviews covered complementary intervention domains: general dietary modification, structured weight-management programs, Mediterranean-diet adherence, diet quality and QoL relationships, and dietetic care delivered in primary care. Table 1 summarizes the scope, methodological features, and principal findings of each review.

### 3.2. Overview of Evidence and Methodological Quality

The methodological rigor of the included reviews was assessed using the AMSTAR-2 instrument (detailed breakdown can be found in the Appendix A). The two Cochrane reviews by Burden et al. (2019) and Shaikh et al. (2020) achieved high methodological confidence, meeting nearly all critical domains [24,25]. Both were prospectively registered, applied duplicate screening and extraction, and used comprehensive search strategies and risk-of-bias assessments. Ryding et al. (2024) also achieved high confidence, having preregistered with PROSPERO, adhered to PRISMA standards, and conducted an AMSTAR-aligned quality appraisal [29]. One of the studies scored moderate confidence, largely due to its reliance on observational data and limited reporting of risk-of-bias integration [27]. Owing to absent protocol registration, incomplete reporting of excluded studies, and insufficient risk-of-bias evaluation, one of the studies ranked low confidence [26]. Collectively, the evidence base is robust in its Cochrane and recent clinical components, though heterogeneity and methodological variability temper the certainty of findings.

In terms of GRADE appraisal, dietetic care (Ryding 2024) [29] demonstrated consistent, reproducible benefits across settings. Weight-management and general dietary counselling have been found to improve nutrition and psychosocial outcomes but vary in intensity and adherence. Mediterranean-diet and QoL-focused reviews were largely observational or small, limiting causal inference. Survival endpoints, long-term adherence, and mechanistic biomarkers remain underexplored. Future large, rigorously designed RCTs are warranted to raise evidence certainty. These findings are summarized in Table 2 and Table 3 and are elaborated upon in the following subsections.

Abbreviations: MD, Mediterranean Diet; QoL, quality of life; SMD, standardized mean difference.

### 3.3. General Dietary Interventions

The Cochrane review by Burden et al. (2019) synthesized twenty-five randomized controlled trials comprising 7259 cancer survivors, approximately 86% of whom were breast cancer patients, who received individualized or group-based dietary counseling [24] (Cochrane Database Syst Rev CD011287). Across trials, participants who received structured dietary interventions reported significantly higher fruit-and-vegetable consumption (+0.41 servings/day, 95% CI 0.10–0.71) and improved diet-quality scores (+3.46 points, 95% CI 1.54–5.38) compared with those in usual-care groups. Modest yet statistically significant reductions in body-mass index (−0.79 kg/m^2^, 95% CI −1.50 to −0.07) were also observed. Although changes in lipid and glucose profiles were inconsistent, the direction of effect generally favored intervention arms. Several trials further demonstrated small-to-moderate improvements in global and physical QoL domains. Broad dietary counseling enhanced nutritional behaviors and modestly reduced adiposity.

### 3.4. Weight Management Interventions

The Cochrane review by Shaikh et al. (2020) focused on overweight or obese breast cancer survivors (BMI ≥ 25 kg/m^2^) and included twenty RCTs involving 2028 participants [25] (Cochrane Database Syst Rev CD012110). These multicomponent programs integrated calorie restriction, physical activity, and behavioral strategies such as cognitive-behavioral therapy or motivational interviewing. Pooled analyses revealed mean weight loss of −2.25 kg (95% CI −3.63 to −0.87), BMI reduction of −1.08 kg/m^2^ (95% CI −1.73 to −0.43), and waist-circumference reduction of −1.73 cm (95% CI −2.56 to −0.90). Participants in structured programs reported significant improvements in overall QoL (SMD 0.74, 95% CI 0.20–1.29), with particularly strong gains among those enrolled in interventions embedding cognitive-behavioral components. No increase in adverse events was noted, and gradual weight loss appeared to preserve lean mass more effectively than rapid restriction. Nonetheless, it was found that intentional, professionally supervised weight reduction is both safe and efficacious in terms of physical and psychological benefits during the survivorship phase.

### 3.5. Mediterranean Diet-Style Intervention

Chen et al. (2023) evaluated adherence to the Mediterranean diet (MD) after breast cancer diagnosis across eleven studies—two RCTs, four cohort studies, and five cross-sectional analyses [27]. Notably, high MD adherence was associated with a 22% reduction in all-cause mortality (HR 0.78, 95% CI 0.66–0.93) and a 33% reduction in non-breast-cancer mortality (HR 0.67, 95% CI 0.50–0.90). Across cohorts, higher MD scores correlated with lower fasting glucose and triglyceride concentrations and with reduced self-reported fatigue, while improvements in physical functioning and vitality were noted in several RCTs. Although the certainty of evidence was rated as low owing to the predominance of observational studies, the consistent direction of effects suggest that MD adherence may support longevity and cardiometabolic health among breast cancer survivors.

### 3.6. Diet Quality and Quality of Life

Barchitta et al. (2020) combined an Italian cross-sectional study of sixty-eight breast cancer survivors with a systematic synthesis of nine interventional trials [26]. Cross-sectionally, higher MD adherence was linked to healthier overall food choices but not to global QoL, indicating that well-being is influenced by multifactorial determinants beyond diet alone. Interventional evidence showed that all diet-based programs, whether low-fat, plant-forward, or MD, yielded improvements in at least one QoL domain, including fatigue, emotional well-being, and body image. The largest benefits occurred when dietary modification was paired with exercise or psychological counseling. These findings suggest an association between diet quality and aspects of QoL, and synergistic multimodal interventions may generate more holistic improvements for cancer survivors.

### 3.7. Dietetic-Led Care in Primary Care Settings

Ryding et al. (2024) examined twelve RCTs encompassing 1138 participants, including 519 breast cancer survivors, evaluating dietitian-led nutritional care delivered in primary-care contexts [29]. Across trials, participants receiving structured dietetic interventions achieved mean weight loss of −3.7 kg and body-fat reduction of −2.3% (*p* < 0.0001). Significant increases in fruit and vegetable consumption and composite diet-quality scores were observed. Programs involving at least three consultations over twelve weeks and incorporating individualized goal-setting achieved the greatest adherence and effectiveness. Two RCTs also demonstrated statistically significant gains in global or functional QoL indices. These data highlight the feasibility and impact of embedding registered dietitians within primary-care survivorship frameworks to promote sustainable nutritional behavior change and durable weight management.

### 3.8. Post-Intervention Maintenance of Dietary Behaviour Changes and Physical Activity

Spark et al. (2012) examined ten randomised controlled trials (RCTs) involving 1536 participants, most of whom were breast cancer survivors [22]. The studies assessed behavioural and clinical outcomes at least three months after the completion of dietary and/or physical activity interventions to evaluate the participants’ maintenance of these changes. Among the four successful trials, only one integrated both physical activity and dietary interventions, with the dietary component being sustained. Notably, the successful trials employed specific targeted strategies that achieved significant improvements in functional status, cancer-related fatigue, and QoL. However, many of the included studies focused solely on physical activity interventions, suggesting that the established benefits of physical activity for post-cancer treatment patients may outweigh those of dietary changes. Further research is needed to explore how clinicians and healthcare professionals can effectively deliver dietary interventions and their potential benefits for cancer survivors.

### 3.9. Role of Nutritional Mobile Apps

Ng et al. (2025) analysed the use of various health applications with nutrition-related functions on participants with cancer or a history of cancer [30]. The thirteen studies included had more than half writing about breast cancer. A combination of logging, tracking, monitoring, provision of dietary information, and coaching was available to participants over a period of eight weeks to six months, during which time a change in nutrition-related health outcomes was measured. Overall, there were significant outcomes, including reduced consumption of high-fat and sugary foods, as well as increased protein and energy intake, among users of the various applications. Some improvements in QoL, body weight, and composition were also identified in participants. Despite the studies being of low to moderate quality, the use of mobile application interventions to maintain dietary interventions seems promising in many settings.

### 3.10. MHBC Interventions on Healthy Eating and Physical Activity

Amireault et al. (2016) aimed to determine the effects of multiple health behaviour change (MHBC) interventions in 4241 participants, mainly comprising breast cancer survivors, across twenty-seven RCTs [23]. In both dietician-led and nurse- or multidisciplinary team-led MHBC interventions, there was a significant improvement in healthy eating, with increased F&V intake and decreased fat intake. These positive changes also corresponded to the increase in diet quality (SMD = 0.37; *p* < 0.0001) but without a change in energy intake.

### 3.11. Lifestyle Interventions with Dietary Strategies

Buro et al. (2024) reviewed sixty-seven studies discussing sixty-one unique lifestyle interventions with dietary strategies, with a particular focus on racial/ethnic diversity among participants [28]. The types of interventions employed, spanning a period of two hours to two years, ranged from in-person classes and meetings to online communication, all aimed at encouraging dietary change. Across all studies, there were significant positive findings in dietary intake and weight-related outcomes, with eighteen addressing diet without physical activity. However, the lack of interventions catered to specific ethnic/racial groups has raised concern over an inadequate understanding of low cancer survival rates in the Black and Hispanic.

## 4. Discussion

To our knowledge, this is the first umbrella review synthesizing dietary and nutrition interventions specifically among breast cancer survivors—exploring how diverse interventions to fill the gap between guideline recommendations and a fragmented underlying evidence base. Based on nine available systematic reviews [21,22,23,24,25,26,27,29,30], it is recognizable that structured nutrition programs, ranging from individualized dietetic counselling and weight-management interventions to adherence to the Mediterranean diet, do produce measurable improvements in diet quality, anthropometric indices, and health-related QoL for breast cancer patients. These benefits extend beyond the acute treatment phase, reinforcing that nutritional optimization constitutes an essential pillar of quality survivorship.

Across all reviews, several consistent patterns emerged. Continuous professional engagement through individualized counseling or structured follow-up was critical for sustaining behavioral adherence. Multicomponent interventions combining diet with physical activity and psychosocial support consistently outperformed dietary programs alone. Mediterranean-style and plant-forward dietary patterns were most strongly associated with reduced mortality and improved metabolic profiles. The use of technology through health applications with nutrition related functions has encouraged patients to do personal tracking and goal setting for their dietary habits. Finally, the inclusion of registered dietitians was identified as a cornerstone for personalizing guidance, reinforcing adherence, and maintaining long-term gains.

Biologically, these findings resonate with mechanistic data linking diet to inflammation, insulin resistance, and estrogen signalling. Diets rich in polyphenols, monounsaturated fats, and fibre reduce systemic inflammatory markers (CRP, IL-6) and improve insulin sensitivity [31]. Such metabolic modulation may mitigate the pro-tumorigenic milieu that underlies recurrence and comorbidity. Behaviourally, structured nutrition interventions foster self-efficacy, agency, and health activation—psychological constructs shown to mediate sustained adherence and QoL gains [32]. These dual biological and behavioural mechanisms position diet as a modifiable determinant not only of survival but also of overall well-being, paralleling the survivorship-care-plan framework advocated by the American Society of Clinical Oncology (ASCO) [33].

These findings are aligned with large cohort evidence from the Women’s Health Initiative and the Nurses’ Health Study, in which higher post-diagnosis dietary quality predicted lower all-cause mortality and improved physical functioning among breast cancer survivors [34]. Global consensus statements including the World Cancer Research Fund/American Institute for Cancer Research (WCRF/AICR) 2018 Continuous Update Project [35] conclude that maintaining a healthy body weight, eating a diet rich in wholegrains, vegetables, fruit, and beans, and limiting alcohol and processed meat are key to preventing recurrence and improving survival. Our findings corroborate and extend these recommendations by demonstrating that dietitian-delivered programs operationalizing these principles yield tangible patient-centered outcomes. As summarized in Table 2 and Table 3, dietician-led models of care consistently reduced weight and body-fat percentage while improving QoL. Furthermore, they echo growing recognition that survivorship is a chronic phase requiring sustained self-management support akin to other non-communicable diseases.

Incorporating nutrition into survivorship care plans demands system-level action. Dietitians should be embedded within multidisciplinary oncology teams and primary-care networks, ensuring continuity from hospital to community. Reimbursement and referral frameworks must support repeated counselling sessions, as frequency and intensity strongly predict adherence [36]. Education for oncologists and primary-care clinicians is equally critical, enabling consistent reinforcement of dietary guidance during routine visits. At a population level, policy efforts should promote food environments conducive to support individuals on plant-based, minimally processed diets [37]. This is crucial as based on a survey of 438 breast cancer survivors in Singapore, many expressed concerns about cancer treatment and recurrence risk, fear of recurrence, and long-term treatment effects across the survivorship trajectory [37].

Nonetheless, from an evidentiary standpoint, several caveats warrant consideration. First, although the Cochrane reviews ensured methodological robustness, most included trials were of short duration (<12 months) and susceptible to bias, including attrition bias. Second, self-reported dietary adherence introduces measurement error; few studies corroborated intake through biomarkers. Third, only a minority of trials stratified outcomes by tumor subtype, treatment regimen, or menopausal status, limiting interpretability across heterogeneous survivor profiles. Finally, the predominance of studies from high-income countries constrains external validity to low- and middle-income settings, where nutritional transitions and food insecurity may shape survivorship differently [38]. These limitations highlight the urgent need for pragmatic, longer-term RCTs and implementation studies that integrate objective nutritional metrics and diverse populations.

For clinicians and survivors, the key takeaway is that structured, dietitician supported and Mediterranean-style dietary approaches produce modest but meaningful improvements in body weight, diet quality and QoL, with emerging evidence suggesting potential benefits for long-term health and survival.

Future research should move beyond efficacy toward implementation and sustainability. Hybrid effectiveness–implementation trials could evaluate the integration of tele-nutrition, mobile health applications, and behavioural-economics “nudges” to sustain engagement [39]. Comparative studies across cultural contexts are needed to refine dietary recommendations for diverse cuisines and socioeconomic settings. Importantly, survivorship research should adopt composite endpoints encompassing both biomedical outcomes (e.g., recurrence, metabolic risk) and experiential outcomes (QoL, social participation), reflecting the multidimensional construct of “quality survivorship”.

## 5. Conclusions

In summary, contemporary evidence indicates that structured, dietitian-supported nutrition programs and adherence to Mediterranean-style dietary patterns improve diet quality, facilitate sustainable weight management, enhance psychosocial well-being, and may contribute to improved long-term health outcomes among breast cancer survivors. Embedding professional nutrition care within survivorship pathways is therefore not an adjunctive measure but a core component of comprehensive, equitable, and high-quality post-cancer care. Future research should continue to refine the evidence base through rigorous, longitudinal, and culturally contextualized studies that strengthen causal inference and guide clinical implementation.

## Figures and Tables

**Figure 1 nutrients-18-00030-f001:**
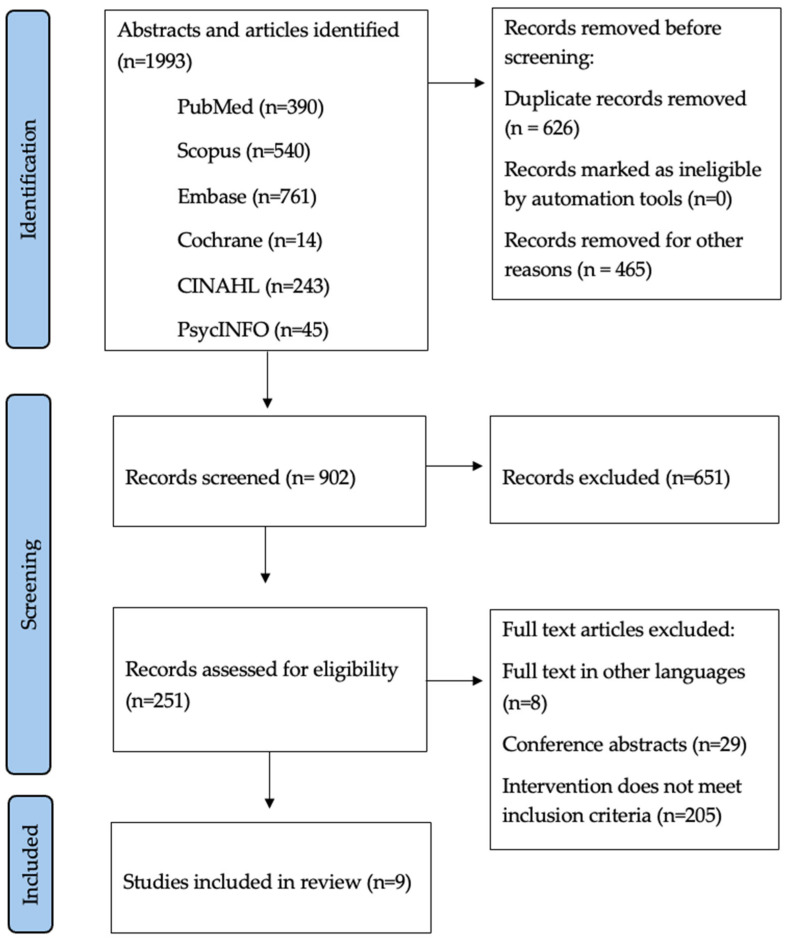
PRISMA flowchart showing the study selection process.

**Table 1 nutrients-18-00030-t001:** Summary of included systematic reviews in current umbrella review.

Author (Year)	Population (*n*)	Intervention Focus	Designs	Key Outcomes	Main Findings
Spark 2012 [22]	10 RCTs (1536 survivors)	Physical activity and dietary behaviour change	RCT	Physical activity, diet, functional status, QoL, cancer-related fatigue	↑ functional status, ↓ cancer-related fatigue, ↑ QoL
Amireault 2016 [23]	33 studies	MHBC intervention	Mixed	F&V intake, fat or energy intake,	↑ healthy eating
Burden 2019 [24]	25 RCTs (>7000 survivors)	General diet counseling	RCT	Diet quality, BMI, QoL	↑ F&V intake, ↓ BMI, ↑ QoL
Shaikh 2020 [25]	20 RCTs (2028 obese survivors)	Weight-management (diet + exercise)	RCT	Weight, BMI, QoL	−2.25 kg weight, ↑ QoL
Barchitta 2020 [26]	68 + 9 trials	Diet quality & QoL	Mixed	QoL domains	Diet + exercise → ↑ QoL
Chen 2023 [27]	11 studies (2 RCT, 4 cohort, 5 cross-sec)	Mediterranean diet	Mixed	Mortality, QoL	↓ all-cause mortality (HR 0.78)
Buro 2024 [28]	61 (41 RCT,	Physical activity and diet	Mixed	Weight, Diet	Change in diet → ↓ weight
Ryding 2024 [29]	12 RCTs (1138 survivors)	Dietitian-led primary care	RCT	Weight, body fat, QoL	−3.7 kg weight, −2.3% fat, ↑ QoL
Ng 2025 [30]	13 studies (7 RCT, 4 single-arm, pretest-postest, 2 quasi-experimental)	Physical activity and diet	Mixed	QoL, nutritional status, dietary intake, body weight and composition	↑ nutritional care, ↑ QoL

Abbreviations: BMI, body mass index; F&V, fruits and vegetables; RCT, randomized controlled trial; QoL, quality of life.

**Table 2 nutrients-18-00030-t002:** GRADE Evidence Summary by Intervention Category.

Intervention Category	Certainty of Evidence (GRADE)	Summary of Evidence (Source Reviews)
General dietary counselling	Moderate (⊕⊕⊕O)	Consistent ↑ diet quality (+3.46 points) [26], ↑ F&V intake (+0.41 servings/day) [26], modest ↓ BMI (−0.79 kg/m^2^); small QoL benefits [26]. Strongest evidence from Cochrane review [26].
Weight-management programs (diet ± PA ± behavioural)	Moderate (⊕⊕⊕O)	Robust reductions in weight (−2.25 kg) [24], BMI (−1.08 kg/m^2^) [24], waist (−1.73 cm) [24]; moderate ↑ QoL (SMD ~0.74) [24]. Intervention intensity and behavioural components improve outcomes.
Mediterranean diet adherence	Low (⊕⊕OO)	High MD adherence → ↓ all-cause mortality (HR 0.78) [21] and ↓ non-BC mortality (HR 0.67) [21]. Improvements in fatigue/metabolic markers; limited RCTs reduce certainty.
Diet quality/nutrition behaviour interventions	Low (⊕⊕OO)	Interventions consistently ↑ at least one QoL subscale (fatigue, emotional, body image). Evidence mixed due to heterogeneity of QoL instruments and small sample sizes.
Dietitian-led primary care nutrition care	High (⊕⊕⊕⊕)	High-quality RCTs show ↓ weight (−3.7 kg) [25], ↓ body-fat % (−2.3%) [25], ↑ diet quality, and ↑ QoL. Consistent across all included trials.
Mobile nutrition apps	Low–Moderate (⊕⊕OO/⊕⊕⊕O)	↓ high-fat/sugar foods, ↑ protein/energy intake, some QoL and weight benefits. Evidence limited by heterogeneity in app functions and small trials [22].
Multiple health behaviour change (MHBC)	Moderate (⊕⊕⊕O)	↑ F&V intake, ↓ fat intake, ↑ diet quality (SMD 0.37). No significant change in energy intake. Behavioural reinforcement critical to effect maintenance [30].

Abbreviations: MD, Mediterranean Diet; QoL, quality of life; SMD, standardized mean difference.

**Table 3 nutrients-18-00030-t003:** GRADE Evidence Summary by Outcome-Level Effect Sizes.

Outcome Category	Outcome	Effect Size (Pooled/Reported)	Direction of Effect	Source Review(s)
1. Body Weight & Composition	Weight change	−2.25 kg (95% CI −3.63 to −0.87)	↓ weight	[24]
	BMI change	−1.08 kg/m^2^ (95% CI −1.73 to −0.43)	↓ BMI	[24]
	Waist circumference	−1.73 cm (95% CI −2.56 to −0.90)	↓ central adiposity	[24]
	Body-fat percentage	−2.3% (*p* < 0.0001)	↓ body fat	[29]
2. Dietary Intake, Diet Quality & Nutrition Behaviours	F&V intake	+0.41 servings/day	↑ intake	[26]
	Diet quality	+3.46 points (95% CI 1.54–5.38)	↑ diet quality	[26]
	Fat intake	↓ fat intake	Improved dietary pattern	[30]
	High-fat/high-sugar food intake	Reduced	↓ unhealthy intake	[22]
3. Quality of Life (QoL)	Global QoL	SMD 0.74 (95% CI 0.20–1.29)	↑ QoL	[24]
	Fatigue	Improved fatigue scores	↓ fatigue	[21,27,29]
	Emotional well-being/Body image	Improved	↑ psychosocial functioning	[27]
4. Other Clinical Outcomes	All-cause mortality	HR 0.78 (95% CI 0.66–0.93)	↓ mortality	[21]
	Non-BC mortality	HR 0.67 (95% CI 0.50–0.90)	↓ non-BC mortality	[21]
	Metabolic markers	Improved glucose, triglycerides	↑ metabolic health	[21]
	Functional status	Improved	↑ function	[25,29]
	Diet adherence maintenance	Mixed	Variable durability	[21,25,29]

Abbreviations: MD, Mediterranean Diet; QoL, quality of life; SMD, standardized mean difference.

## Data Availability

This is a review of published studies; no new data were generated in this study.

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
