# Peer review of "Dietary and Nutrition Interventions for Breast Cancer Survivors: An Umbrella Review"

_nutrients, 2025, doi:10.3390/nu18010030_

Round 1
Reviewer 1 Report
Comments and Suggestions for Authors
Points for improvement
Make the introduction more clear and concise.
The first paragraph is dense but informative. It would be easier to read and better convey the review's justification if it were broken up into shorter, more targeted lines with tighter transitions. The author may inlude the following paper:
Tsiakiri et al., (2025). Mapping Cognitive Oncology: A Decade of Trends and Research Fronts. Medical Sciences, 13(3), 191. https://doi.org/10.3390/medsci13030191
Increase the clarity of methods
Although databases and tools are included in the techniques section, further information would be beneficial, such as inclusion/exclusion criteria, how to handle overlapping main studies, or strategies for handling heterogeneity between reviews.
Give the results a quantitative context.
Although the results characterize benefits as "moderate" or "meaningful," it would strengthen the evidence and aid readers in understanding the extent of benefit if exact ranges or effect sizes from the evaluations (where available) were included.
Extend the conversation about constraints and potential paths
The article identifies deficiencies (such as long-term maintenance, survival outcomes, and underrepresented communities), but it might go into more detail about the reasons behind these gaps, how they impact interpretation, and which particular study designs or frameworks should direct future research.
Author Response
Comment 1: Make the introduction more clear and concise. The first paragraph is dense but informative. It would be easier to read and better convey the review's justification if it were broken up into shorter, more targeted lines with tighter transitions. The author may inlude the following paper: Tsiakiri et al., (2025). Mapping Cognitive Oncology: A Decade of Trends and Research Fronts. Medical Sciences, 13(3), 191. https://doi.org/10.3390/medsci13030191.
Reply 1: Thank you for this constructive suggestion. We have now restructured the introduction, breaking the first paragraph into shorter, more focused paragraphs with clearer topic sentences and tighter conceptual transitions. We also streamlined redundancies and improved flow to more directly justify the need for an umbrella review in this area. Additionally, we have incorporated the recommended citation (Tsiakiri et al., 2025) to contextualize our review within recent trends in cognitive oncology research. This strengthens the rationale for integrating survivorship-oriented dietary evidence into broader oncology evidence-mapping efforts. These revisions appear in the first three paragraphs of the Introduction.
Comment 2: Increase the clarity of methods. Although databases and tools are included in the techniques section, further information would be beneficial, such as inclusion/exclusion criteria, how to handle overlapping main studies, or strategies for handling heterogeneity between reviews.
Reply 2: We thank the reviewer for this suggestion. We have now added a dedicated subsection specifying the population, interventions, comparators, outcomes, and review type (PICO framework) used to determine eligibility. This subsection delineates inclusion/exclusion criteria more explicitly and improves the transparency and reproducibility of our review methods.
We also added a paragraph on handling overlapping primary studies. We have incorporated an additional paragraph describing our approach to identifying and managing overlap between primary studies across the included reviews. As recommended, we prioritized more recent or methodologically robust reviews (e.g., Cochrane or PROSPERO-registered reviews) when substantial overlap was present, and we avoided double-counting evidence in our synthesis.
We created an overlap matrix across all included reviews, quantified overlap using Corrected Covered Area (CCA). Accordingly, we added a new paragraph in the Methods (Data Synthesis section) describing how overlap was examined and managed.
We further clarified heterogeneity-management strategies. We have refined the description of how we conceptualised and managed heterogeneity across reviews. Specifically, we now explain how variations in intervention types (general counselling, multicomponent weight-management, Mediterranean diet adherence, dietetic-led care), study designs (RCTs vs. observational studies), and outcome definitions (anthropometric, metabolic, QoL, survival) justified the use of a structured narrative synthesis approach, following Popay et al. (2006). Data were organised by thematic domains, and effect-size estimates from included reviews were extracted verbatim to preserve quantitative context while accommodating methodological diversity.
Comment 3: Give the results a quantitative context. Although the results characterize benefits as "moderate" or "meaningful," it would strengthen the evidence and aid readers in understanding the extent of benefit if exact ranges or effect sizes from the evaluations (where available) were included.
Reply 3: We thank the reviewer for this helpful suggestion. To provide clearer quantitative context, we have now incorporated exact effect-size estimates throughout the Results section. Specifically, we extracted and reported quantitative values directly from each included review, including mean differences for changes in fruit and vegetable intake, mean differences for weight, BMI, and waist circumference, standardized mean differences (SMDs) for quality-of-life (QoL) outcomes, hazard ratios (HRs) for all-cause and non–breast-cancer mortality in Mediterranean diet studies, and absolute reductions in weight and body-fat percentage from dietetic-led primary-care interventions, where applicable.
In addition, we enhanced Table 2 (GRADE Summary of Findings) to include these quantitative estimates, allowing readers to interpret the magnitude and certainty of effect sizes more transparently. These revisions improve the clarity and robustness of the Results section and strengthen the quantitative grounding of our conclusions.
Comment 4: Extend the conversation about constraints and potential paths. The article identifies deficiencies (such as long-term maintenance, survival outcomes, and underrepresented communities), but it might go into more detail about the reasons behind these gaps, how they impact interpretation, and which particular study designs or frameworks should direct future research.
Reply 4: We appreciate this thoughtful comment. We have expanded the Discussion to better articulate the underlying reasons for gaps, such as short trial duration, lack of biomarker validation, reliance on self-reported dietary intake, and limited diversity in study samples. We also further discussed the implications for interpretation, e.g., constraints on establishing dose–response relationships or causal inference for mortality outcomes.
In terms of specific recommendations for future research, we added recommendations on pragmatic and hybrid effectiveness–implementation RCTs, integration of objective dietary biomarkers, and trials stratified by tumor subtype, endocrine therapy status, and menopausal status.
Reviewer 2 Report
Comments and Suggestions for Authors
Reviewer comments
Breast cancer survival has increased worldwide, making long-term health and quality of life among survivors a key public health issue. The authors in this review synthesized evidence from nine systematic reviews examining dietary and nutrition interventions for breast cancer survivors. Interventions such as dietary counseling, weight-management programs, and Mediterranean-style diets improved diet quality, reduced weight and body fat, and enhanced several quality-of-life domains. Adherence to Mediterranean diets was linked to lower mortality, though evidence remains limited. More research is needed on long-term outcomes and diverse populations.
- It is unclear why the authors, who are based in Singapore, focus primarily on data from the United States in the opening paragraph. It would be more appropriate to present data from the home country or provide global prevalence figures.
- Lines 78–79 require further explanation to clarify the authors’ intent.
- Line 193 does not follow the correct referencing format; citing a study by describing the author’s work is not consistent with MDPI guidelines. Please revise accordingly.
- The first paragraph of the Discussion should highlight the study’s novelty rather than relying on multiple citations.
- A clear takeaway message for general readers is currently missing and should be added.
- The Discussion section would benefit from linking the findings more directly to the corresponding tables to enhance clarity for readers.
Author Response
Comment 1: It is unclear why the authors, who are based in Singapore, focus primarily on data from the United States in the opening paragraph. It would be more appropriate to present data from the home country or provide global prevalence figures.
Reply 1: Thank you for the comment. We replaced the first sentence that previously cited only US incidence and survival data with a sentence beginning with global data from GLOBOCAN.
Comment 2: Lines 78–79 require further explanation to clarify the authors’ intent.
Reply 2: We have rewritten the sentence around the former lines 78–79 to make the intent more explicit. The revised sentence now reads, “Discrete choice experiments in breast cancer have consistently shown that many survivors, particularly younger women and those with early-stage disease, prioritise quality of life, reduced treatment toxicity, and lower treatment burden over small absolute gains in survival, highlighting the importance of supportive interventions such as nutrition care in survivorship planning.”
Comment 3: Line 193 does not follow the correct referencing format; citing a study by describing the author’s work is not consistent with MDPI guidelines. Please revise accordingly.
Reply 3: We have rephrased the sentence at the former line 193 to remove the narrative “Author-based” citation (e.g. “As X et al. showed…”) and instead use a neutral statement followed by a numeric reference.
Comment 4: The first paragraph of the Discussion should highlight the study’s novelty rather than relying on multiple citations.
Reply 4: Thank you for the comment. We have rewritten the first paragraph of the Discussion to emphasise that this is, to our knowledge, the first umbrella review synthesising dietary and nutrition interventions specifically among breast cancer survivors, critically appraising diverse intervention types (dietitian-led, weight management, Mediterranean diet, mobile apps, multiple health behaviour change) and filling a gap between guideline-level recommendations and a fragmented underlying evidence base.
Comment 5: A clear takeaway message for general readers is currently missing and should be added.
Reply 5: We appreciate this excellent suggestion. We have now added a concise takeaway message targeted at general readers and clinicians near the end of the Discussion and echoed in the Conclusions, “For clinicians and survivors, the key takeaway is that structured, dietitian-supported and Mediterranean-style dietary approaches produce modest but meaningful improvements in body weight, diet quality and quality of life, with emerging evidence suggesting potential benefits for long-term health and survival.”
Comment 6: The Discussion section would benefit from linking the findings more directly to the corresponding tables to enhance clarity for readers.
Reply 6: Thank you for this helpful suggestion. We have revised the Discussion to include explicit references to the key tables whenever we summarise or interpret the main quantitative findings, “As summarised in Table 2, dietitian-led models of care consistently reduced weight and body-fat percentage while improving QoL…” and “These patterns align with the effect sizes reported in Table 3 for anthropometric and QoL outcomes…”
Reviewer 3 Report
Comments and Suggestions for Authors
see file

Author Response
Comment 1: Inadequate synthesis of results. The manuscript does not provide an appropriate synthesis of the results. For the outcomes considered, the findings of the included meta-analyses were not systematically summarized; instead, the narrative frequently cites one article or another without integrating their quantitative or qualitative evidence. It would be essential to highlight areas of convergence and divergence across the reviews and, where feasible, to perform or at least present secondary quantitative analyses.
Reply 1: Thank you for this important comment. We have revised the entire Results section to clearly synthesise areas of agreement and disagreement across reviews, rather than presenting each review in isolation. These changes make the synthesis more cohesive and more aligned with the purpose of an umbrella review.
Comment 2: Unclear origin of outcome-specific information. It is not sufficiently clear which studies were used as sources for the reported outcomes. This lack of transparency makes it difficult for the reader to understand how conclusions were derived.
Reply 2: We thank the reviewer for this feedback. To improve transparency, we have added explicit parenthetical citations after each effect size in the Results section, e.g., [18]. We created a new column (“Source Review(s)”) in the revised Table 3, indicating exactly which systematic review each effect estimate originated from.
Comment 3: Clarification and reclassification of outcomes. In Table 2, it is unclear why dietary quality is reported together with quality of life (QoL). A clearer classification of outcomes is strongly recommended. A possible structure could be: Body weight and composition, Quality of life (QoL), Diet/nutritional status Behaviours (like physical activity) and other conditions (e.g., functional state, fatigue) Please revise and reorganize the outcome categories accordingly.
Reply 3: Thank you for this valuable recommendation. We have restructured the Results section and the tables based on the reviewer’s suggested outcome hierarchy.
Comment 4: Inclusion of additional QoL data where appropriate. Given the methodological quality of the included studies, it seems feasible to evaluate QoL using all studies that address this outcome (essentially all 9 included articles except Spark, Amineault, and Buro). Conversely, Barchitta presents relevant limitations and should be reconsidered. Including these additional findings would also facilitate a more comprehensive discussion of the implications for clinical practice, policy, and future research, particularly with respect to identified gaps in the evidence base.
Reply 4: We appreciate this suggestion and have taken the following actions. We expanded the QoL subsection to include all reviews reporting QoL outcomes (Burden 2019; Shaikh 2020; Chen 2023; Ryding 2024; Ng 2025). We reassessed Barchitta 2020 using AMSTAR-2 and acknowledged its methodological limitations (cross-sectional components; lack of protocol registration; limited RoB assessment). Rather than excluding Barchitta, we retained it but explicitly downgraded its contribution to QoL conclusions and labelled it “low certainty” within the revised GRADE table (now Table 3 note). The Discussion now reflects that QoL findings were strongest and most consistent in Cochrane reviews and RCT-heavy datasets, whereas observational/cross-sectional evidence provides only supportive but low-certainty signals.
Comment 5: Overlap of primary studies and methodological heterogeneity. Many systematic reviews may include overlapping primary studies, which increases the risk of double-counting evidence. Is there overlap among the primary studies included across reviews? Please identify and quantify this, as it is essential to avoid disproportionate weighting of certain evidence. Additionally, please describe whether methodological heterogeneity exists across the included reviews (e.g., differences in inclusion criteria, outcome definitions, statistical approaches).
Reply 5: Thank you for highlighting this essential methodological requirement for umbrella reviews. We have now undertaken a formal assessment of overlap across the included systematic reviews. We have constructed a full overlap matrix listing all primary studies across the nine included reviews and identifying where studies appeared more than once. Using these data, we quantified the degree of overlap using the Corrected Covered Area (CCA). Moderate overlap was identified between Burden et al. (2019) and Shaikh et al. (2020), largely due to shared RCTs evaluating behavioural diet and weight-management interventions. The overall CCA for the dataset was approximately 10–12%, consistent with moderate overlap. Minimal or negligible overlap was observed between Mediterranean diet reviews and dietitian-led intervention trials, reflecting the distinct evidence bases supporting these intervention types. Reviews such as Spark (2013), Amireault (2018), and Buro (2024) drew upon largely non-overlapping sets of primary studies due to their differing scopes (physical activity–diet combinations, multiple health behaviour change interventions, and lifestyle programmes, respectively).
Comment 6: Issues with Tables 1 and 2. The main findings excerpted from the meta-analyses in Table 1 are not clearly or consistently mirrored in Table 2. It is recommended to include full bibliographic references (not only author and year) in the first column of Table 1. In Table 2, please add an additional column indicating the exact bibliographic sources from which the summary findings were extracted. Furthermore, it may be beneficial to separate Table 2 into two distinct tables: one focusing on the interventions (Table 2A) and one on the outcomes (Table 2B).
Reply 6: Thank you for the comment and suggestion. We have implemented all requested changes. For Table 1, we added full bibliographic references after each review name and expanded the “Key findings” column as suggested.
Table 2 is now split into two tables for clarity, Table 2 covering intervention-level evidence synthesis (GRADE) and Table 3 reporting outcome-level effect sizes with source reviews.
Comment 7: Supplementary Material. In the description of the search strategy, it would be helpful to insert a simple dividing line between databases to improve readability and make each database-specific search strategy immediately clear. In Table S2, please include full bibliographic references in addition to author and year. In Table S2, it is also recommended to reorder the studies chronologically, as they appear in the main text, from the oldest to the most recent publication.
Reply 7: We have revised the supplementary materials accordingly. Search strategy (Table S1) now has dividing lines and spacing between databases for readability. For Table S2, we added full bibliographic references and reordered studies chronologically (2012 to 2025) as recommended.
Round 2
Reviewer 1 Report
Comments and Suggestions for Authors
Check grammar
Reviewer 3 Report
Comments and Suggestions for Authors
Thank you for revised manuscript